# Dose-dependent response to infection with SARS-CoV-2 in the ferret model and evidence of protective immunity

Kathryn A. Ryan [1], Kevin R. Bewley [1], Susan A. Fotheringham[1], Gillian S. Slack[1], Phillip Brown [1], Yper Hall[1], Nadina I. Wand [1], Anthony C. Marriott [1], Breeze E. Cavell [1], Julia A. Tree[1], Lauren Allen[1], Marilyn J. Aram[1], Thomas J. Bean[1], Emily Brunt[1], Karen R. Buttigieg[1], Daniel P. Carter [1], Rebecca Cobb[1], Naomi S. Coombes[1], Steve J. Findlay-Wilson[1], Kerry J. Godwin[1], Karen E. Gooch[1], Jade Gouriet[1], Rachel Halkerston[1], Debbie J. Harris[1], Thomas H. Hender[1], Holly E. Humphries [1], Laura Hunter[1], Catherine M. K. Ho[1], Chelsea L. Kennard[1], Stephanie Leung[1], Stephanie Longet [1], Didier Ngabo [1], Karen L. Osman [1], Jemma Paterson[1], Elizabeth J. Penn[1], Steven T. Pullan[1], Emma Rayner[1], Oliver Skinner[1], Kimberley Steeds[1], Irene Taylor[1], Tom Tipton [1], Stephen Thomas[1], Carrie Turner[1], Robert J. Watson[1], Nathan R. Wiblin[1], Sue Charlton[1], Bassam Hallis[1], Julian A. Hiscox[2], Simon Funnell [1], Mike J. Dennis[1], Catherine J. Whittaker[1], Michael G. Catton[3], Julian Druce[3], Francisco J. Salguero[1] & Miles W. Carroll[1,4 ✉]

There is a vital need for authentic COVID-19 animal models to enable the pre-clinical evaluation of candidate vaccines and therapeutics. Here we report a dose titration study of SARS-CoV-2 in the ferret model. After a high ($5 \times 10^6$ pfu) and medium ($5 \times 10^4$ pfu) dose of virus is delivered, intranasally, viral RNA shedding in the upper respiratory tract (URT) is observed in 6/6 animals, however, only 1/6 ferrets show similar signs after low dose ($5 \times 10^2$ pfu) challenge. Following sequential culls pathological signs of mild multifocal bronchopneumonia in approximately 5–15% of the lung is seen on day 3, in high and medium dosed groups. Ferrets re-challenged, after virus shedding ceased, are fully protected from acute lung pathology. The endpoints of URT viral RNA replication & distinct lung pathology are observed most consistently in the high dose group. This ferret model of SARS-CoV-2 infection presents a mild clinical disease.

[1] National Infection Service, Public Health England (PHE), Porton Down, Salisbury, Wiltshire SP4 0JG, UK. [2] Institute of Infection and Global Health, University of Liverpool, Liverpool L69 2BE, UK. [3] Victorian Infectious Diseases Reference Laboratory, Peter Doherty Institute for Infection and Immunity, Melbourne, Victoria 3000, Australia. [4] Nuffield Department of Medicine, Oxford University, Oxford OX1 3SY, UK. ✉email: miles.carroll@phe.gov.uk

Coronaviruses are positive sense, single stranded RNA viruses belonging to the family Coronaviridae[1]. These viruses can infect a range of animals, including humans and usually cause a mild respiratory infection, much like the common cold. Two highly pathogenic coronaviruses have emerged in the human population in the last 20 years; severe acute respiratory syndrome (SARS-CoV) and middle eastern respiratory syndrome (MERS-CoV). SARS-CoV infected approximately 8000 people worldwide with a case fatality rate (CFR) of 10%, while MERS-CoV has infected approximately 2500 people with a CFR of 36%[2].

In December 2019 several pneumonia cases of unknown cause emerged. Deep sequencing analysis from lower respiratory tract samples from patients indicated the cause to be a novel coronavirus[3]. The causative agent of this novel coronavirus disease (COVID-19) was identified as SARS-CoV-2. Globally, as of 23 August 2020, there have been 23,057,288 confirmed cases of COVID-19, including 800,906 deaths, reported to WHO[4]. The global mortality rate is yet to be determined because there are concerns about the variability between countries of reporting rates, health care provision, socioeconomic and innate genetic differences. Approximately 80% of patients display only mild symptoms, with approximately 14% displaying severe symptoms such as dyspnoea and low blood oxygen saturation. Around 6% of cases become critical, with respiratory failure, septic shock and/or multiple organ failure[5]. There is an urgent need to develop suitable animal models to evaluate antivirals, therapeutics and vaccine candidates against SARS-CoV-2.

Ferrets have been used extensively to model the disease caused by influenza virus[6–12] infection as well as human RSV[13,14], mumps virus[15], Ebola virus[16,17] and Nipah virus[18,19]. Due to the presence of a compatible form of ACE2, the virus receptor, on cells of the ferret respiratory tract, these animals were developed an as effective model for SARS-CoV[20–23]. The ferret has been shown to shed detectable virus from the URT as well as exhibiting comparable clinical symptoms associated with milder cases of the infection[21] and has shown similar pathology in the lung to that observed in humans[22]. SARS-CoV-2 spike protein has been shown to exhibit many similarities in its amino acid sequence and protein structure to the receptor binding domain of SARS-CoV[24] and also utilises ACE2 for cell entry[25], suggesting ferrets would be a suitable host for a model for COVID-19.

In this study, our aim is to understand if ferrets are a suitable species for a model of human SARS-CoV-2 infection. Animals are challenged intranasally with a range of titres of SARS-CoV-2 ($5 \times 10^2$, $5 \times 10^4$ and $5 \times 10^6$ pfu) in 1 ml volume. We report that high and medium dose challenge induce URT RNA shedding, associated with LRT tissue viral RNA and lung pathology. Re-challenge of recovered ferrets show a reduced URT viral shedding and lung pathology. This model aids our understanding of SARS-CoV-2 pathogenesis and naturally acquired immunity, additionally, it provides a pre-clinical model for the evaluation of co-infections, vaccines and therapeutics.

## Results

**Study design**. Ferrets were challenged intranasally with Victoria/1/2020[26] SARS-CoV-2, in 1 ml volume to increase the incidence of virus reaching the lung[9], at three different titres representing a high, medium and low dose (Table 1). A high titre stock of challenge virus was prepared (passage 3), and quality control sequencing showed it was identical to the original stock received from the Doherty Institute and did not contain a commonly reported 8 amino acid deletion in the furin cleavage site[27]. Following the initial challenge, a re-challenge with the high dose ($5 \times 10^6$ PFU) was performed on day 28 post-challenge (pc). The four (two per group) remaining ferrets in medium and low groups were re-challenged via the same, intranasal, route using a 1 ml volume alongside a control group of two naïve control ferrets (group 5).

**Viral shedding following challenge**. Viral RNA was detected in the nasal wash of 6/6 ferrets in the high dose group from day 1 pc and continued to be detected at varying levels until day 20 pc (Fig. 1a). The peak in viral RNA shedding was seen between days 2 and 4 pc for all ferrets in the high dose group. Following a decline in viral RNA (2/2 animals) to below the limit of quantification of the assay at day 13 pc, an increase was seen at days 16 and 18 pc. Both Group 1 survivors were euthanised on day 21 pc at which point no viral RNA was detected in their nasal washes. Live virus was isolated in the nasal wash of 2/6 of the high dose ferrets at 3 and 4 dpc (Fig. 1a) at 30 and 110 pfu/ml, respectively. This is in line with the viral subgenomic RNA (sgRNA) detected in the nasal wash (Fig. 1b), where these two samples, along with eight additional samples, are quantifiable for sgRNA.

In the medium dose group 6/6 ferrets also had detectable viral RNA in nasal washes from day 1 pc. The peak of viral RNA shedding was more variable in the medium dose group, with some ferrets peaking at days 2–3 pc (4/6) and others peaking at days 5–6 pc (2/6). A decline was then seen until day 11 pc where viral RNA levels fell below the limit of quantification, but viral RNA was still above the limit of detection of the assay. By day 16 no more viral RNA was detected. Quantifiable viral RNA was only found in the nasal wash of 1/6 ferrets in the low challenge dose group. This ferret was euthanised on day 5 pc. No other ferrets in the low dose group were found to shed quantifiable viral RNA in their nasal wash. Additionally, live virus was not detected

**Table 1 Experimental animal groups.**

| Group | | Number of animals | Initial challenge virus titre (pfu/ml) | Re-challenge virus titre (pfu/ml) | Euthanasia/challenge days |
|---|---|---|---|---|---|
| 1 | High | 6 | $5 \times 10^6$ | | 1 Ferret euthanised at day 3, 5, 7, 14 pc; 2 ferrets euthanised day at 21 pc |
| 2 | Medium | 6 | $5 \times 10^4$ | $5 \times 10^6$ | 1 Ferret euthanised at days 3, 5, 7, 14 pc; 2 ferrets re-challenged at day 28 pc; 1 ferret euthanised at days 33 & 36 pc |
| 3 | Low | 6 | $5 \times 10^2$ | $5 \times 10^6$ | 1 Ferret euthanised at days 3, 5, 7, 14 pc; 2 ferrets re-challenged at day 28 pc; 1 ferret euthanised at days 33 & 36 pc |
| 4 | Naïve sentinel | 2 | PBS | | 2 Ferrets euthanised at day 20 pc |
| 5 | Naïve control | 2 | | $5 \times 10^6$ | 1 Ferret euthanised at days 33 & 36 pc |

A total of 22 ferrets were distributed across 5 groups. All inoculations were performed intranasally with 1 ml of fluid. pc, post-challenge.

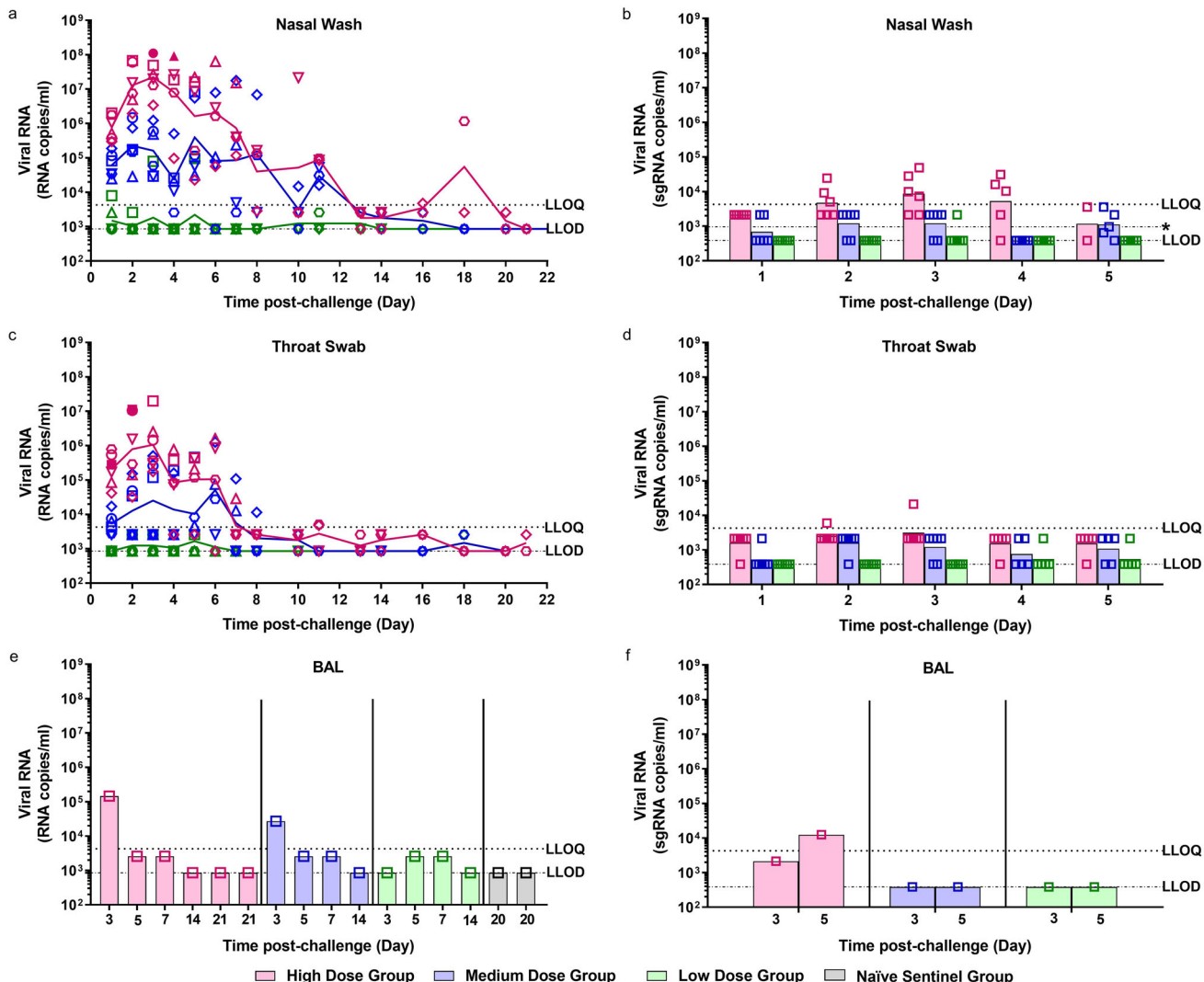

**Fig. 1 Viral RNA Shedding.** Nasal washes and swabs were collected at days 1 to 8, 10, 11, 13, 14, 16, 18 and 20 pc for all virus challenged groups. Viral genomic RNA was quantified by RT-qPCR at all timepoints. No viral RNA was detected in any samples taken from the naïve sentinel ferrets. Samples that were found to be positive by plaque assay are represented by a solid shape. (**a**) Nasal washes (**c**) Throat swabs (**e**) Bronchoalveolar lavage (BAL) collected at necropsy (numbers indicate day post-challenge the ferret was euthanised). Symbols show values for individual animals, lines represented the calculated group geometric means. The presence of viral subgenomic RNA was assessed in (**b**) nasal washes, (**d**) throat swabs and (**f**) BAL from 1 to 5 dpc. Bars show geometric mean for each group and symbols show values for each individual animal. n = 6 ferrets per group with numbers decreasing by 1 at 3, 5, 7 and 14 dpc. The dashed horizontal lines show the lower limit of quantification (LLOQ) and the lower limit of detection (LLOD). [*LLOD range for nasal washes represents 2 undetected samples at 5 dpc had <5 µl template tested].

in the nasal wash of the medium and low dose groups and sgRNA analysis showed that sgRNA was detected above the level of detection of the assay but below the level of quantitation.

A similar trend in the titre of viral RNA detected in nasal wash samples was observed in the throat swab samples during the first week after challenge (Fig. 1c). The amount of viral RNA detected in the throat swab samples of ferrets in the high dose group (6/6) peaked at day 3 pc. In contrast, however, detection of viral RNA in throat swab samples was less prolonged than in the nasal passage, with no quantifiable viral RNA detected past day 11 pc. Live virus was isolated in the throat swabs of 1/6 ferrets on 2 consecutive days, 240 and 50 pfu/ml, respectively. This is in line with the sgRNA quantified in the throat swabs (Fig. 1d).

Illumina sequencing of nasal wash RNA extracts showed little variation between the genomes isolated at days 5 and 6 pc and the original sequence of the virus inoculated into the ferrets. Only one non-synonymous SNP was identified, in the day 5 pc for a

ferret from the medium dose group; a T2152I mutation within the orf1ab polyprotein, no further timepoints were collected for this animal as it was euthanised at day 5 pc.

Viral RNA was detected at quantifiable levels in the bronchoalveolar lavage (BAL) of each ferret euthanised (scheduled) on day 3 pc from the high dose (1/1) and medium dose (1/1) groups (Fig. 1e). Viral RNA was detected above the level of detection of the assay but below the level of quantitation for ferrets across all three challenge groups at day 5 and 7 pc. sgRNA was detected in the BAL at day 5 pc for the high dose group (Fig. 1f) but not in any of the medium dose animals. There was no viral RNA detected in the BAL of any of the other ferrets after scheduled euthanasia.

Viral RNA was detected in the nasal cavity (Fig. 2a) of individual high dose ferrets euthanised at 3, 5 and 7 dpc, with the highest amount of viral RNA detected in the ferret at 7 dpc. Importantly, sgRNA detected in the nasal cavity mirrors the

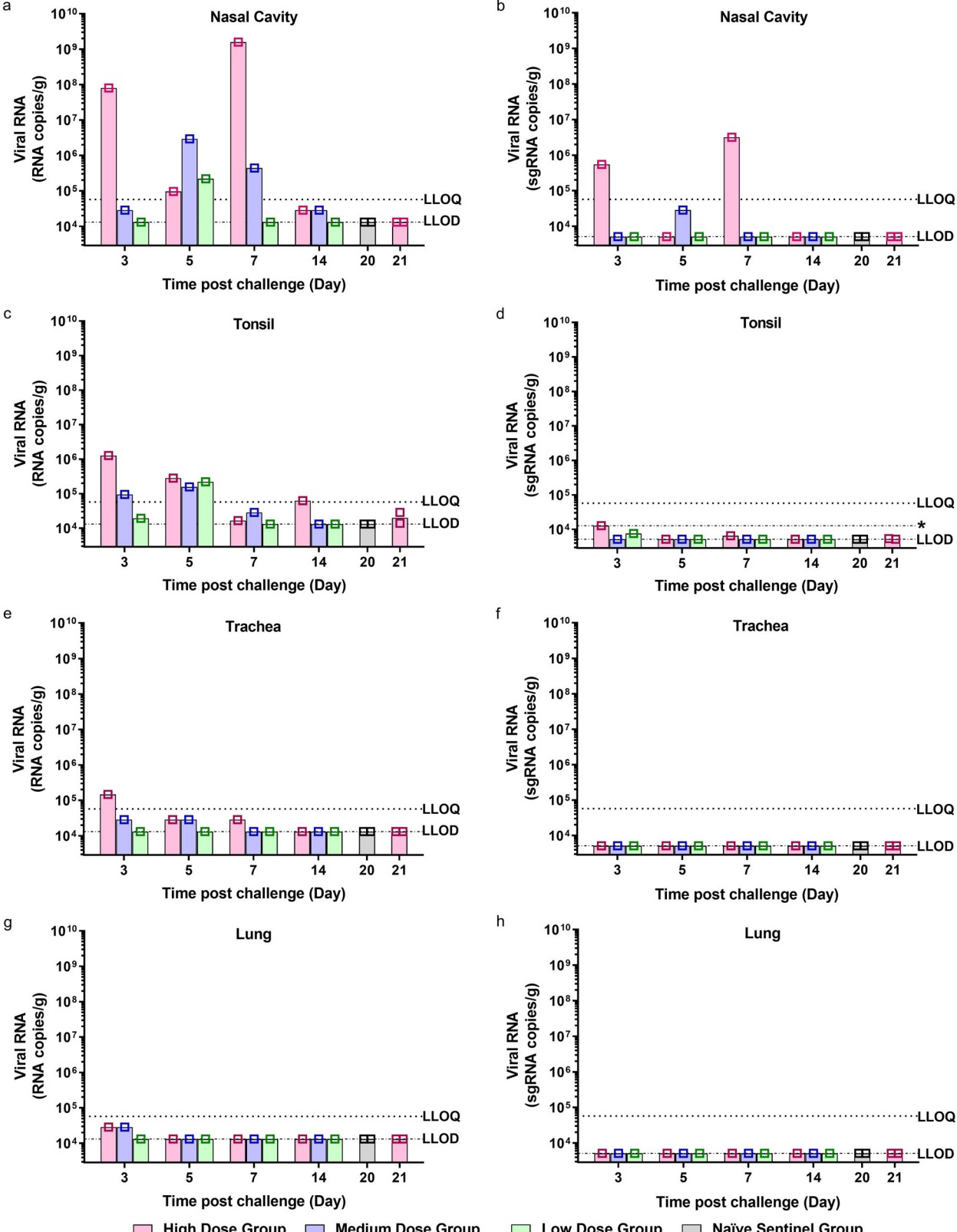

**Fig. 2 Viral RNA Shedding in Tissues. a**, **b** Nasal cavity, **c**, **d** tonsil, **e**, **f** trachea and **g**, **h** lung were collected at euthanasia timepoints (days 3, 5, 7, 14, 20 and 21 pc) for all groups. $n = 6$ ferrets per group ($n = 2$ ferrets in the naive sentinel group), with one ferret from each group culled at each timepoint (except day 20 and 21 pc where $n = 2$ ferrets from the naive sentinel and high dose groups were culled respectively). Viral genomic RNA (**a**, **c**, **e** and **f**) and viral subgenomic RNA (**b**, **d**, **f** and **h**) was quantified by RT-qPCR. Bars show values for individual animals. The dashed horizontal lines show the lower limit of quantification (LLOQ) and the lower limit of detection (LLOD). *LLOD range for tonsil represents undetected samples homogenised in smaller volume.

genomic RNA findings, however at a much lower level (Fig. 2b). Viral RNA was also detected in individual ferrets challenged with the medium dose culled at 3, 5 and 7 dpc. At 14 dpc viral RNA was detected above the level of detection of the assay but below the level of quantitation in the high and medium dose ferrets. The low dose ferret that was shown to shed viral RNA in its nasal wash had a positive virus signal in nasal cavity and tonsil (Fig. 2c) tissue at 5 dpc when it was culled. Viral RNA was also quantified in the tonsils (Fig. 2c) in high and medium dose ferrets at 3 and 5 dpc, with no detectable sgRNA found in the tonsils (Fig. 2d).

Viral RNA was quantified in the trachea (Fig. 2e) of one of the ferrets euthanised at 3 dpc; and detected in high and medium dose ferrets across 3, 5 and 7 dpc. Viral RNA was detected above the level of detection of the assay but below the level of quantitation in the lungs (Fig. 2g) of the high and medium dose ferrets culled at 3 dpc. Viral RNA was not detected in the lungs of any other ferrets euthanised at any other timepoints. Viral sgRNA was not detected in the trachea (Fig. 2f) or lungs (Fig. 2h) of any ferrets. No viral RNA was detected in the liver (Supplementary Fig. 1b), jejunum (Supplementary Fig. 1c) or colon (Supplementary Fig. 1d) of any of the ferrets, from any of the groups.

Detection of viral RNA in the rectal swabs was found to be variable across the different dose groups (Supplementary Fig. 1a). The highest viral RNA load was observed in a ferret in the high dose group but there was a less consistent pattern of RNA detection which did not continue past day 7 pc. In the medium dose group, 4/6 ferrets were found to have detectable viral RNA in their rectal swabs between days 2 and 8 pc. No viral RNA was detected in any of the rectal swabs collected from the low dose group following challenge.

**Clinical signs**. The normalised summed incidence of clinical scores for each group of ferrets is shown in Fig. 3a and total summed scores are shown in Table 2. At day 9 pc all 3/3 ferrets in the high dose group showed reduced activity, a similar observation was made in the medium dose group but later, on day 10 pc. Reduced activity was accompanied by ruffled fur, a sign that the ferrets were not grooming regularly. By day 14 pc ferrets in the medium dose group stopped showing signs of reduced activity and by day 15 pc the high dose groups stopped showing signs of reduced activity. Ferrets in the high dose group had the highest normalised cumulative clinical score (summed across all time points) (14.01), followed by the medium dose group (6.99) with sporadic instances recorded in the low dose group. No fever (±1 °C from baseline at least two consecutive occasions) was detected in any ferret, in any group (Fig. 3b); instead body temperature remained within the normal range. No weight loss was observed in any ferret in any group, below baseline; however, the SARS-CoV-2 infected ferrets failed to gain as much weight as the ferrets in the control (phosphate buffered saline, PBS) group, although this difference was not statistically significant (Fig. 3c).

**Histopathology**. The nasal cavity from high dose ferrets showed a minimal to mild necrosis of epithelial cells and mild inflammatory cell infiltration (Fig. 4a) from days 3 to 7 pc. However, abundant epithelial cells from the nasal cavity were stained for viral RNA at day 3 pc (Fig. 4b). Occasional scattered cells expressing viral RNA were observed in high dose animals at days 5 and 7 pc and medium dose animals at days 3, 5 and 7 pc. Similarly, very few scattered epithelial cells were stained for viral RNA in the trachea and larynx from high and medium dose animals at days 3, 5 and 7 pc.

No remarkable gross lesions were observed in the infected animals. Upon histological examination of the lungs of ferrets from the high and medium dose groups, a mild multifocal bronchopneumonia from days 3 to 14 pc was observed. Mild necrosis of the bronchiolar epithelial cells was observed together with inflammatory cell infiltration of neutrophils and mononuclear cells within the bronchiolar luminae, mostly affecting animals from the high dose group at days 3, 5 and 7 pc (Fig. 4c). This bronchopneumonia was characterised by the infiltration of inflammatory cells, mostly neutrophils, but also macrophages and lymphocytes, in approximately 10–15% of the lung section at day 3 pc decreasing to less than 5% at days 5 and 7 pc. The medium dose group showed mild bronchopneumonia in less than 5% of the lung sections at days 3 and 5 pc, while only occasional infiltration was observed in animals from the low dose group. Bronchiolar inflammatory infiltration was not observed in control (PBS) animals (Fig. 4h). Few cells stained positive for viral RNA using in situ hybridisation (RNAScope). Few type I and occasionally type II pneumocytes and alveolar macrophages were positive for viral RNA at days 3, 5 and 7 pc (Fig. 4d) in high and middle dose animals. The presence of viral RNA was not associated with histopathological lesions. Occasionally, mild proliferation of bronchus-associated lymphoid tissue (BALT) was observed surrounding damaged bronchi and bronchioles at the early stages of the disease, with slightly more severity at days 14 and 21 pc (high dose) (Fig. 4e). Mild interstitial pneumonia with an increase in the thickness of the interalveolar septa was observed from day 3 pc towards the end of the experiment in high and medium dose groups (Fig. 4e). Ferrets from the high dose group showed mild proliferation of type II pneumocytes from day 7 pc onwards (Fig. 4e).

The liver showed multiple foci of inflammatory cell infiltration in the portal areas, composed of mainly macrophages, lymphocytes and occasional plasma cells (Fig. 4f). This multifocal infiltration was more severe in animals from the high and medium dose groups from day 3 pc, compared to the low dose group or control (PBS) animals (Fig. 4i), which only showed minimal presence of portal inflammation. No other remarkable changes were observed in any other tissue. However, occasional positive cells (absorbing epithelial enterocytes and goblet cells) were also observed in the small and large intestine from high and medium dose at days 3, 5 and 7 pc, not associated with histopathological lesions (Fig. 4g).

**Immune response to SARS-CoV-2 infection**. Neutralising antibody titres for ferrets infected in the high dose and medium dose groups generally increased longitudinally following challenge as illustrated in Supplementary Fig. 2a. The average fold increase of neutralising antibodies from days 8 to 14 pc was about the same for both the high and medium dose groups. The low dose group had comparatively low neutralising antibodies throughout the time course. Further analysis of ferret sera showed that the presence of SARS-CoV-2 spike specific IgG (Supplementary Fig. 2b) demonstrated a similar trend, with titres increasing over time in the high and medium dose groups. Upon euthanasia, lung mononucleocytes (MNCs) were isolated from the high dose ferrets at 14 and 21 dpc, the medium dose ferret at 14 dpc, and the naïve sentinel ferrets culled at 20 dpc. Cellular immune responses were assessed in lung MNCs using whole live SARS-CoV-2 to stimulate the production of IFNγ (Supplementary Fig. 3). SARS-CoV-2 specific immune responses were seen in the high and medium dose ferrets euthanised at 14 and 21 dpc.

One naïve sentinel group ferret was shown to have neutralising antibodies to SARS-CoV-2 as well as the presence of SARS-CoV-2 spike specific IgG upon euthanasia at day 20 pc. This ferret showed no clinical signs of SARS-CoV-2 infection and samples (nasal wash and swabs) taken at baseline, day 11 and day 20 pc were shown to be PCR negative for SARS-CoV-2. Furthermore

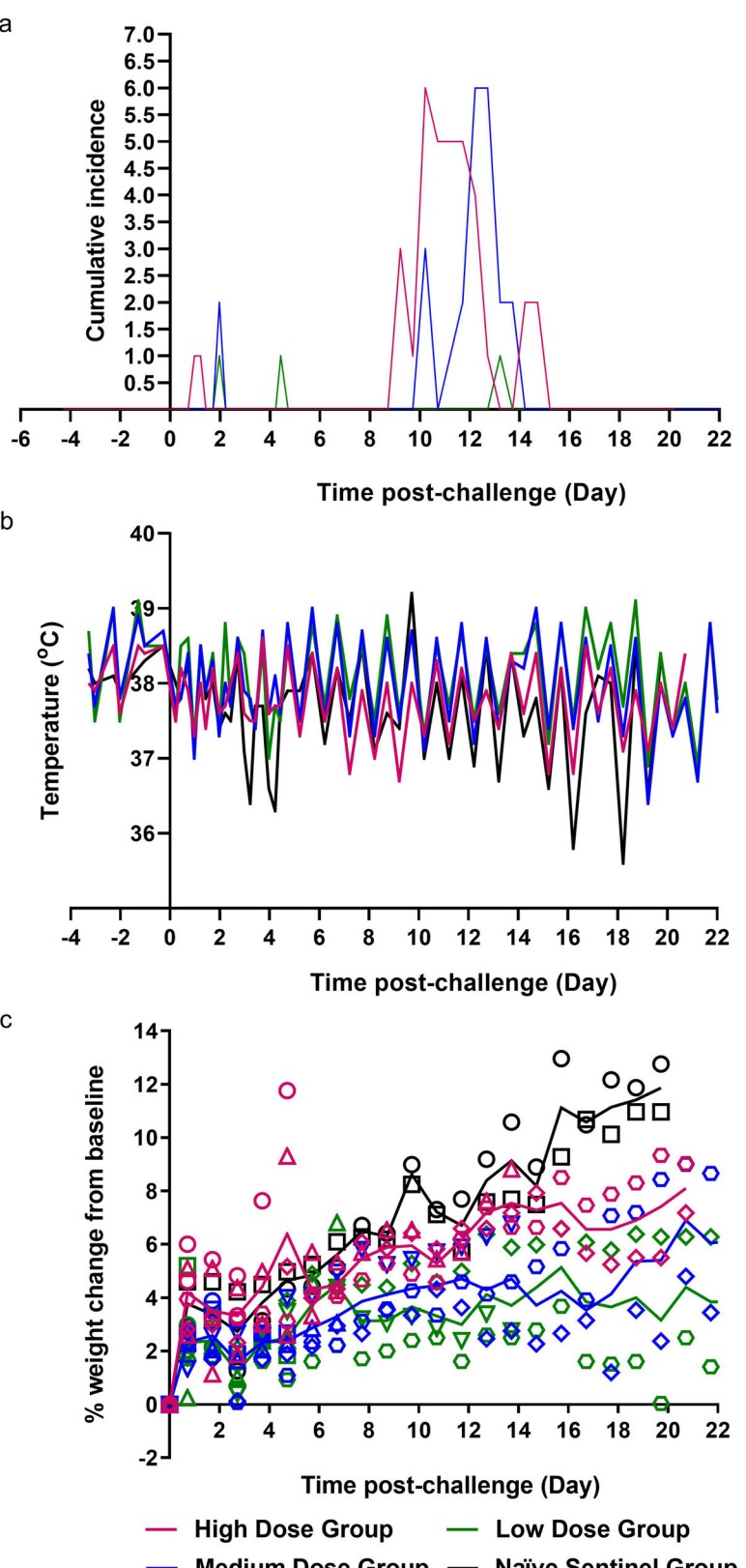

there was no evidence of pathology in any of the tissues taken from both naïve sentinel ferrets euthanised on day 20 pc. Following euthanasia, viral RNA was detected (but not quantified) in tissue in the colon of this ferret. Viral RNA was not detected in any other tissues analysed (nasal cavity, tonsil, trachea, lung, liver and jejunum). Interestingly, the cellular immune response seen in the lung MNCs (Supplementary Fig. 3) of the naïve sentinel ferret showed a high SARS-CoV-2 specific immune response to whole live SARS-CoV-2, paralleling the detection of neutralising antibodies and SARS-CoV-2 spike specific IgG. These results suggest that this ferret could have inadvertently become infected with SARS-CoV-2.

**Fig. 3 Clinical Observations. a** Clinical observations were carried out four times daily (approximately 6 h apart) for the first 5 days and then twice daily (approximately 8 h apart) for the remaining time. Observations were summed for each group of ferrets. **b** Temperatures were taken at the same time as clinical observations, using the identifier chip, to ensure any peak of fever was recorded. Mean temperatures are displayed on the graph. **c** Weight was recorded daily and percentage weight change from baseline was plotted. Points show values for individual animals, lines represented the calculated group means. $n = 6$ ferrets per group ($n = 2$ ferrets in the naive sentinel group) with numbers decreasing by 1 at 3, 5, 7 and 14 dpc. The table illustrates the summed scores for each clinical observation noted for each of the groups during specific days post-challenge and post re-challenge. Activity in ferrets was scored as follows; 0 = alert and playful, 1 = alert, playful when stimulated, 2 = alert, not playful when stimulated, 3 = not alert or playful. Ruffled fur was given a score of 1. Activity scores of 1 were given to ferrets during the initial challenge. Upon re-challenge ferret activity was recorded as 1 or 2 indicating increased lethargy in ferrets following re-challenge.

**Table 2 Clinical scores.**

| Group | Clinical scores (individual instances summed) | | | | | |
| | Initial challenge | Day 0–7 pc | Day 8–14 pc | Day 15–21 pc | Re-challenge | Day 0–8 post re-challenge |
|---|---|---|---|---|---|---|
| 1 High | 1 Ferret euthanised at day 3, 5, 7, 14 pc; 2 ferrets euthanised at day 21 pc | Activity =1; 2 | Activity =1; 19 Ruffled fur; 12 | Activity =1; 2 | | |
| 2 Medium | 1 Ferret euthanised at day 3, 5, 7, 14 pc | Activity =1; 2 | Activity =1; 12 Ruffled fur; 8 | 0 | 1 Ferret euthanised at day 5 & 8 post re-challenge | Activity =1; 4 Activity =2; 2 Ruffled fur; 1 |
| 3 Low | 1 Ferret euthanised at day 3, 5, 7, 14 pc | Activity =1; 1 Sneeze; 1 | Activity =1; 1 | 0 | 1 Ferret euthanised at day 5 & 8 post re-challenge | Activity =1; 2 Activity =2; 1 |
| 4 Naïve sentinel | 2 Ferrets euthanised at day 20 pc | 0 | 0 | 0 | | |
| 5 Naïve control | | | | | 1 Ferret euthanised at day 5 & 8 post challenge | Activity =1; 2 |

**Re-challenge of ferrets with high dose SARS-Cov-2 results in absence of lung pathology.** Four previously infected ferrets, two from the medium and low dose challenge groups, had neutralising titres of 1:274, 1:250, 1:82 and 1:55 at day 26 pc, respectively. At day 28 pc, these ferrets and two naïve control animals were challenged intranasally with the high dose of SARS-Cov-2 ($5 \times 10^6$ pfu). Though shedding of viral RNA from the nasal wash was similar in all groups on day 2 post re-challenge, viral RNA levels subsequently decreased in the previously challenged animals ($n = 4$), with the medium dose group showing rapid decrease to below quantifiable levels by day 5 post re-challenge. Viral RNA levels in the nasal wash continued to stay above quantifiable levels in the challenged naïve control group, although they began to fall at day 8 pc (Fig. 5a). Similar results were seen in the throat swab and rectal swabs (Supplementary Fig. 4a, b), with reduced viral shedding seen in the re-challenged animals. Viral RNA was only detected in the BAL of the ferret from the naïve control group euthanised at 5 dpc (Supplementary Fig. 4c). Subgenomic RNA was detected but not quantified in nasal wash (Supplementary Fig. 5a) and throat swabs (Supplementary Fig. 5b) for re challenged animals and animals challenged for the first time.

Analysis of viral RNA in the tissues clearly show reduced viral RNA present in the nasal cavity and tonsil in the medium dose re-challenged group at 5 days post re-challenge compared to the low dose re-challenged and naïve challenged control group (Fig. 5b). Analysis of viral subgenomic RNA present in the nasal cavity showed a reduction of the amount of RNA present in the medium group compared to the low and naïve control groups at day 5 post re-challenge (Supplementary Fig. 5c).

Animals in the re-challenged medium and low dose groups exhibited weight loss from baseline that was not seen at initial challenge for any of the animals in any of the challenge groups (Fig. 5c). Re-challenged animals also experienced increased clinical observations of lethargy and ruffled fur that was not observed at such an early stage in the initial challenge (Table 2). In contrast, the two naïve control animals did not experience weight loss below baseline after infection and they did not suffer the same level of clinical observation as the re-challenged animals (Fig. 5c).

The cellular immune response in the lungs of a low dose (Group 3) re-challenge ferret and a naïve control (Group 5) ferret at day 36 (8 days post re-challenge, respectively) were compared. Supplementary Fig. 6 shows SARS-CoV-2 specific cellular immune responses, as determined by IFN-γ ELISpot. The number of secreting cells detected after re-stimulation of lung MNCs with peptide pools spanning the spike protein varied between ferrets from each group. The strongest response is detected in the re-challenge ferret after ex vivo re-stimulation with whole live virus. Upon histological examination the upper and lower respiratory tracts from animals in both re-challenged groups showed no remarkable lesions and an absence of significant bronchopneumonia (Fig. 5d, e), that was observed in ferrets initially challenged with $5 \times 10^6$ pfu for the first time, i.e. the original high dose ferret or the naïve control infected group included for the 're-challenge' (Fig. 5f). This parallels the absence of pathology observed in the two naïve sentinel ferrets euthanised at day 20 pc.

## Discussion

This study demonstrates ferrets are susceptible to experimental intranasal infection with a low passage isolate of SARS-CoV-2 strain Victoria 1[28]. A high dose ($5 \times 10^6$ pfu/ml in 1 ml volume) intranasal challenge in ferrets produced mild clinical signs, consistent lung pathology and a viral shedding pattern that aligns with the mild to moderate disease seen in clinical cases of COVID-19[28].

Previously published SARS-CoV-1 challenge studies conducted in the ferret show that a lower dose of ($10^3$ TCID$_{50}$) is sufficient

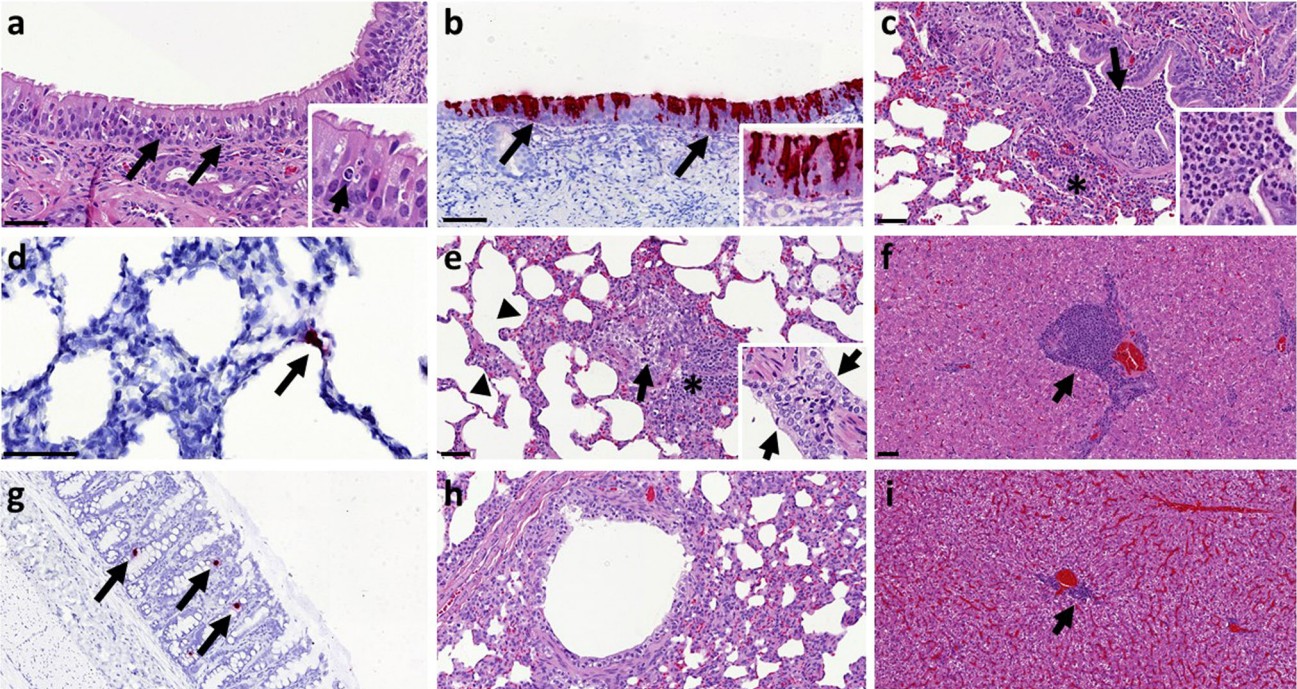

**Fig. 4 Histopathological findings and presence of SARS-CoV-2 RNA in tissue sections from ferrets inoculated with SARS-CoV-2.** Two sections from each tissue or organ were evaluated independently by two qualified pathologists and representative images are shown. **a** Nasal cavity, day 3 pc, Group 1, H&E staining. Mild epithelial cell necrosis (arrows) and minimal inflammatory cell infiltration within the epithelium; bar = 50 µm; inset = close up image of an inflammatory cell within the epithelial layer. **b** Nasal cavity, day 3 pc, Group 1, SARS-CoV-2 viral RNA detection (RNASCope staining). Presence of viral RNA in abundant epithelial and sustentacular cells from the nasal cavity mucosa; bar = 50 µm; inset = close up image showing abundant viral RNA within the olfactory epithelium. **c** Lung, day 5 pc, Group 1, H&E staining. Moderate bronchopneumonia with neutrophil and macrophage inflammatory infiltrate within the bronchiolar lumina (arrow). Mild peribronchiolar infiltration of mononuclear cells (*); bar = 50 µm; inset = clse up image of the bronchiolar inflammatory cell infiltration showing abundant neutrophils. **d** Lung, day 3 pc, Group 2, SARS-CoV-2 viral RNA detection (RNASCope staining). Presence of viral RNA in type II pneumocyte (arrow); bar = 50 µm. **e** Lung, day 21 pc, Group 1, H&E staining. A Bronchiole with mild inflammatory infiltration in the lumina (arrow) and attenuation of the epithelial cells. Moderate peribronchiolar infiltration of mononuclear cells (*) and mild interalveolar septal inflammatory cell infiltration with thickening of the wall (arrowheads); bar = 50 µm; inset = close up image of mils pneumocyte II proliferation in the lung at day 7 pc. **f** Liver, day 21 pc, Group 1, H&E staining. Moderate multifocal hepatitis with mononuclear cell infiltration in the portal areas (arrow); bar = 50 µm. **g** Colon, day 3 pc, Group 1, SARS-CoV-2 viral RNA detection (RNASCope staining). Presence of viral RNA in scattered cell within the absorptive epithelium (arrows); bar = 50µm. **h** Lung, naïve animal, H&E staining. Bronchiole showing no inflammatory reaction; bar = 50 µm. **i** Liver, naïve animal, H&E staining. Minimal hepatitis with mononuclear cell infiltration in the portal areas (arrow); bar = 50 µm.

to cause a mild disease in the ferret[20,21]. We have shown that a high ($5 \times 10^6$ pfu) and medium ($5 \times 10^4$ pfu) dose intranasal challenge of SARS-CoV-2 results in an infection characterised by prolonged viral RNA shedding in all ferrets (days 1–11 pc), accompanied by observable clinical signs from day 8 pc for both high and medium dose groups. Onset of clinical symptoms were delayed by approximately 24 h in the medium dose animals. Both doses also induced classical pathology of bronchial pneumonia involving 10% and 3% of recipient lungs, respectively. A low dose intranasal challenge of the same SARS-CoV-2 virus ($5 \times 10^2$ pfu) appeared to result in infection of only one ferret which shed detectable viral RNA in the upper respiratory tract (UTR) but failed to show any remarkable lesions in the respiratory tract.

In the high and medium dose groups, virus was readily detected using in situ hybridisation in the upper respiratory tract of ferrets, with a peak at 3 dpc. These findings aligned with the detected shedding of viral RNA and sgRNA from nasal washes and throat swabs which also peaked at day 3–4 pc and detection of viral and sgRNA in the upper respiratory tract tissue This upper respiratory infection mirrors the clinical disease recently reported in mild cases of humans infected with SARS-CoV-2 infection[28].

Recent reports indicate that COVID-19 patients appear to shed viral RNA intermittently after recovery from disease with some individuals being tested and found to be positive again after

release from isolation[29]. This report is in alignment with the observations in the two ferrets challenged with the high dose of SARS-CoV-2 (euthanised at day 21 pc) which appeared to continue to shed detectable viral RNA from the upper respiratory tract up to day 18 pc even though these animals had developed neutralising antibodies.

The main histopathological finding in approximately 10% of the lung tissue sections in the high dose group consisted mainly of a multifocal bronchiolitis, with inflammatory infiltrates within the airways and some alveolar species. This finding is similar, but less severe, to the findings in the published reports about SARS-CoV-1 ferret challenge models[22,30,31]. In this study, mild alveolar damage was observed in the acute phase. At later time points, mild proliferation of type II pneumocytes, with interstitial infiltrates and peribronchiolar cuffing, was recoded, consistent with evolution from the acute phase.

Mild to moderate multifocal hepatic inflammatory cell infiltration has been widely reported in viral infections in animals, and has been previously described in SARS-CoV-1 infected ferrets[32]. However, the periportal infiltrates may not be associated with injury to the surrounding tissue and they are reported as a common background finding in laboratory ferret species. The presence of infected enterocytes has been reported for SARS-CoV-1 and SARS-CoV-2 in humans[33] and different ferret models[34,35].

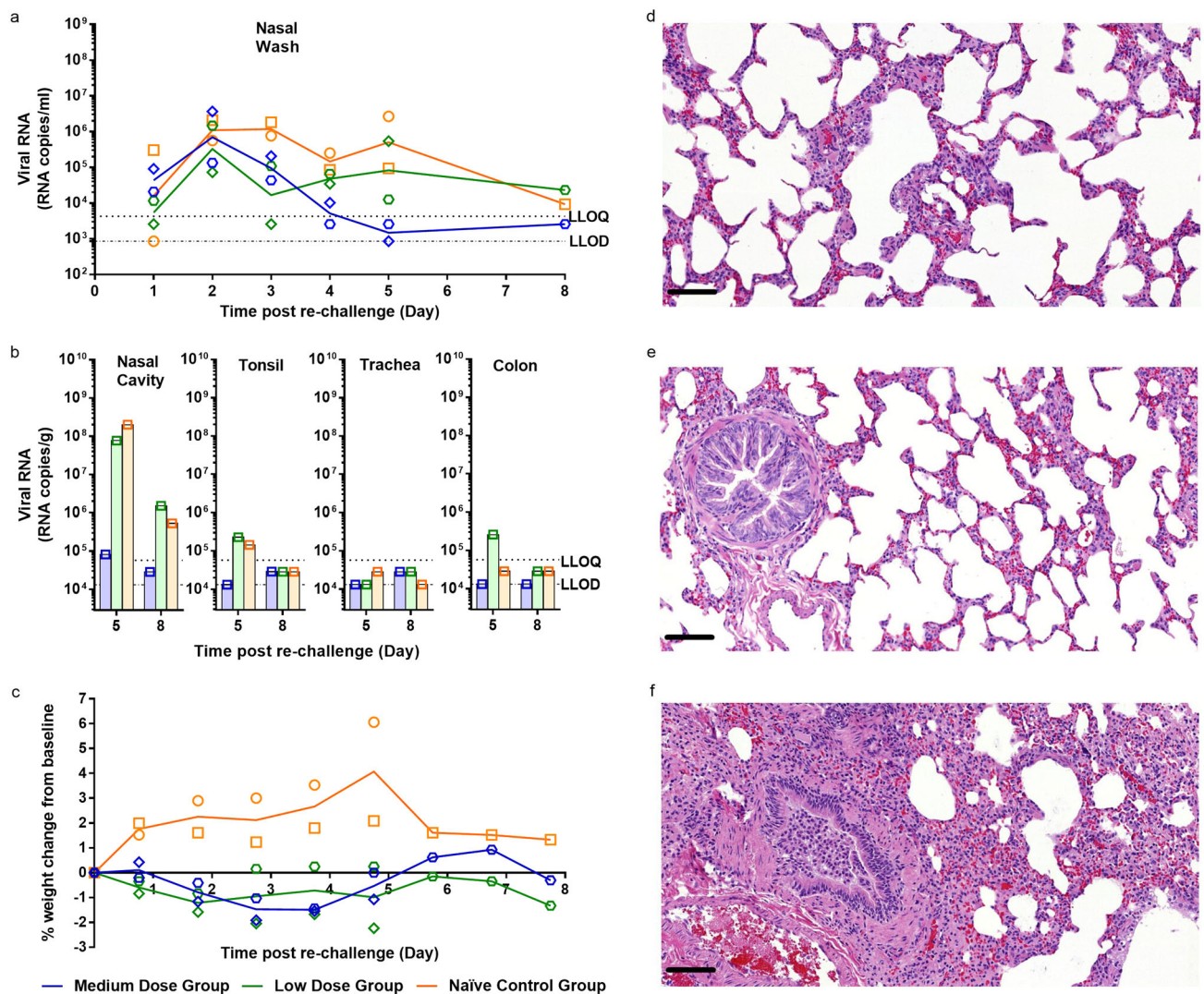

**Fig. 5 Re-challenge of ferrets with SARS-CoV-2. a** Nasal washes were collected at days 1–5 post re-challenge (days 29–33 post-original challenge). Viral RNA was quantified by RT-qPCR. **b** Nasal cavity, tonsil, trachea and colon were collected at euthanasia timepoints (days 5 and 8 post re-challenge) for all groups. Viral RNA was quantified by RT-qPCR. Bars show values for individual animals. The dashed horizontal lines show the lower limit of quantification (LLOQ) and the lower limit of detection (LLOD). **c** Percentage weight change from baseline. Baseline was calculated as average of the two most recent weights taken preceding re-challenge. *n* = 2 ferrets per group with numbers decreasing by 1 at 5 dpc. **d** Medium dose re-challenged ferret at day 5 post re-challenge. No remarkable changes in alveoli or terminal bronchiole. **e** Low dose re-challenged ferret at day 5 post re-challenge. No remarkable changes in alveoli or bronchiole. **f** Control group ferret challenged for the first time (day 28 pc). Inflammatory infiltration within bronchiolar lumen and mild infiltration of alveolar septa; lesions comparable with those observed in the original high dose group. Bar = 50 μm.

The upper respiratory virus replication, reported here, in the high and medium dose groups of animals, support the observations of Shi et al.[34] and Kim et al.[35] which found peak URT viral RNA shedding between days 4 and 6 pc. Shi et al.[34] also reported mild lung pathology associated with SARS-CoV-2 infection similar to our medium dose animals, but this was not as extensive as that seen in our high dose challenge group ferrets.

Shi et al.[34], Kim et al.[35] and Richard et al.[36] report live virus isolation from RNA positive nasal wash samples. In this study, live virus was detected in a minority of nasal washes and throat swabs, even though high levels of viral RNA were detected. However, analysis of viral subgenomic RNA confirmed that a minority of samples contained replicating virus, this aligned with samples in which live virus was detected. A possible reason for this observation during this study could be how samples were processed following collection. Nasal washes, throat swabs and

BAL were pelleted prior to sampling for RNA extraction and live virus assays. It is feasible that virus recovered from animals remained strongly cell-associated and was depleted by centrifugation. Alternatively, this result may have accurately reflected low levels of viable virus presence which others have reported even though viral RNA can be detected. For human swabs and sputum samples, it has been noted that infectious virus was never recovered from samples with a viral RNA load of less than $10^6$ copies/ml[28].

Neutralising antibody levels developed in ferrets in the medium and high dose challenge groups within 14 days. While the levels of serum neutralising antibodies did not increase in the low dose animals, mucosal, humoral and cellular immunity could have played a role during re-challenge. Accepting the small number of animals in our re-challenge study, the finding that medium dosed animals displayed reduced viral RNA shedding in the URT, and

both low and medium dosed animals showed an absence of lung pathology following re-challenge is encouraging; it suggests that there may be potential benefits of naturally acquired immunity and is in line with the observation reported by Bao et al.[37] in which previously infected rhesus macaques were protected against re-challenge with SARS-CoV-2.

SARS-CoV-2 spike protein-specific immune responses seen in a low dose re-challenged ferret were compared to that of a primary challenge ferret. This comparison showed that the response to the virus appears to be higher on re-challenge. However, ferrets challenged with our high dose of SARS-CoV-2 displayed increased clinical observations and lost weight from baseline following re-challenge, hinting at enhanced disease but a larger study would be required to effectively assess this observation. Alternatively, these clinical signs may be a perfectly normal host response to infection in a pre-immune individual whilst the immune system is successfully clearing a large challenge dose.

In addition to the ferret, hamsters and non-human primates (NHPs) have also been developed as models of SARS-CoV-2. The NHP model appears to exhibit a mild clinical disease much like the ferret but has an increased incidence of lung pathology. However, the Syrian hamster model has been found to exhibit weight loss following SARS-CoV-2 infection and the virus appears to replicate efficiently in the lungs causing severe pathological lesions commonly reported in COVID-19 patients with pneumonia[38,39]. The ferret has been well characterised for other respiratory viruses and provides a useful comparative model for the assessment of SARS-CoV-2 pathogenicity and transmission, in addition to the evaluation of vaccines, drugs and therapeutics. Access to practical volumes of blood, in the ferret model, for sequential in-life samples and the availability of an immunological tool kit are also advantages.

This study demonstrates that ferrets challenged with $5 \times 10^6$ pfu or $5 \times 10^4$ pfu displayed only mild clinical signs of SARS-CoV-2 infection. As seen in the clinical setting, pathological features appear to be less severe than those reported after ferrets were infected with SARS-CoV-1[21,31].

This ferret model of intranasal SARS-CoV-2 infection presents three key measurable endpoints: (a) consistent URT viral RNA shedding; (b) detectable lung pathology; and (c) post viral fatigue. Reductions in URT RNA shedding during the first 14 days post intranasal challenge could be an attractive indicator of the efficacy of candidate therapeutics and vaccines. It may be wise, however, to euthanise prior to 14 dpc to more accurately assess the impact on lung pathology especially when looking for signs of vaccine-enhanced disease[20,40,41]. We believe the high dose intranasal challenge will provide the most distinct disease endpoints. However, with its reduced level of lung pathology, the medium dose challenge may provide a higher level of sensitivity to some interventions, as was observed when assessing therapeutics to influenza in the ferret model[42].

## Methods

**Viruses and cells**. SARS-CoV-2 Victoria/01/2020[26] was generously provided by The Doherty Institute, Melbourne, Australia, at P1 after primary growth in Vero/hSLAM cells and subsequently passaged twice at PHE Porton in Vero/hSLAM cells [ECACC 04091501]. Infection of cells was with ~0.0005 MOI of virus and harvested at day 4 by a single freeze thaw cycle and clarification by centrifugation at $1000g$ for 10 min. Whole genome sequencing was performed, on the P3 challenge stock, using both Nanopore and Illumina as described previously[43]. Virus titre of the challenge stocks was determined by plaque assay on Vero/E6 cells [ECACC 85020206]. Cell lines were obtained from the European Collection of Authenticated Cell Cultures (ECACC) PHE, Porton Down, UK. Cell cultures were maintained at 37 ºC in MEM (Life Technologies, California, USA) supplemented with 10% foetal bovine serum (Sigma, Dorset, UK) and 25 mM HEPES (Life Technologies). In addition, Vero/hSLAM cultures were supplemented with 0.4 mg/ml of geneticin (Invitrogen) to maintain the expression plasmid.

**Animals**. Twenty-two healthy, female ferrets (*Mustela putorius furo*) aged 7 months were obtained from a UK Home Office accredited supplier (Highgate Farm, UK). The mean weight at the time of challenge was 1032 g/ferret (range 870–1239 g). Animals were housed in pairs at Advisory Committee on Dangerous Pathogens (ACDP) containment level 3. Cages met with the UK Home Office 'Code of Practice for the Housing and Care of Animals Bred, Supplied or Used for Scientific Procedures' (December 2014). Access to food and water was *ad libitum* and environmental enrichment was provided. All experimental work was conducted under the authority of a UK Home Office approved project licence that had been subject to local ethical review at PHE Porton Down by the Animal Welfare and Ethical Review Body (AWERB) as required by the 'Home Office Animals (Scientific Procedures) Act 1986'.

**Experimental design**. Before the start of the experiment animals were randomly assigned to challenge groups, to minimise bias. The weight distribution of the animals was tested to ensure there was no statistically significant difference between groups (one-way ANOVA, $p > 0.05$). An identifier chip (Bio-Thermo Identichip, Animalcare Ltd, UK) was inserted subcutaneously into the dorsal cervical region of each animal. Prior to challenge animals were sedated by intramuscular injection of ketamine/xylazine (17.9 and 3.6 mg/kg bodyweight). Challenge virus was delivered by intranasal instillation (1.0 ml total, 0.5 ml per nostril) diluted in phosphate buffered saline (PBS).

Three different doses of virus were delivered to three groups ($n = 6$) of ferrets: high ($5 \times 10^6$ pfu/ml), medium ($5 \times 10^4$ pfu/ml) and a low ($5 \times 10^2$ pfu/ml) dose. For the high, medium and low dose groups, individual ferrets were scheduled for euthanasia on day 3 ($n = 1$), day 5 ($n = 1$), day 7 ($n = 1$) and day 14 ($n = 1$). For the high dose group, the remaining 2 ferrets were euthanised on day 21 ($n = 2$). The mock-infected animals ($n = 2$) received an intranasal instillation of sterile PBS and were euthanised on day 20.

On day 28 pc the remaining ferrets in the low ($n = 2$) and medium ($n = 2$) groups were re-challenged with $5 \times 10^6$ pfu by the intranasal route. Additional naïve control ferrets ($n = 2$) were also challenged on day 28, to provide a re-challenge control. All 6 animals were monitored for clinical signs and one ferret from each group was euthanised on day 33 and the remaining animals were euthanised on day 36.

Nasal washes, throat and rectal swabs were taken at days −1, 1–8, 10, 11, 13, 14, 16, 18 and 20 pc. They were also taken at days 1–5 and 8 post re challenge (days 29–33 and 36 pc). Whole blood and serum were collected at 2, 5, 8, 11 and 14 dpc for all ferrets. Whole blood and serum were collected at days 2, 5 and 8 (days 30, 33 and 36 pc) post re-challenge for all remaining ferrets. The negative control ferrets ($n = 2$) had nasal washes, throat swabs, whole blood and serum taken at −1 and 11 dpc. At necropsy nasal washes, throat and rectal swabs, whole blood and serum were taken alongside tissue samples for histopathology. Nasal washes were obtained by flushing the nasal cavity with 2 ml PBS. For throat swabs, a flocked swab (MWE Medical Wire, Corsham, UK) was gently stroked across the back of the pharynx in the tonsillar area. Throat and rectal swabs were processed, and aliquots stored in viral transport media (VTM) and AVL at −80 °C until assay.

**Clinical and euthanasia observations**. Animals were monitored for clinical signs of disease four times daily (approximately 6 h apart) for the first 5 dpc and then twice daily (approximately 8 h apart) for the remaining time. Clinical signs of disease were assigned a score based upon the following criteria. Activity was scored as follows: 0 = alert and playful, 1 = alert, playful when stimulated, 2 = alert, not playful when stimulated, 3 = not alert or playful. Ruffled fur was given a score of 1. No other clinical signs were noted. In order to meet the requirement of the project license, immobility, neurological signs or a sudden drop in temperature were automatic euthanasia criteria. Animals were also deemed to have reached a humane endpoint if their body weight was at or below 30% baseline. If any ferret reached any of these three euthanasia criteria, they were to be immediately euthanised using a UK Home Office approved Schedule 1 procedure. However, no animals reached these end-points during this study.

Temperature was taken using a microchip reader and implanted temperature/ID chip. Temperature was recorded at each clinical scoring point using the chip to ensure any peak of fever was recorded. Animals were weighed at the same time each day from the day before infection until euthanasia.

**Necropsy procedures**. Ferrets were anaesthetised with ketamine/xylazine (17.9 and 3.6 mg/kg bodyweight) and exsanguination was effected via cardiac puncture, followed by injection of an anaesthetic overdose (sodium pentabarbitone Dole-lethal, Vetquinol UK Ltd, 140 mg/kg). A necropsy was performed immediately after confirmation of death. The BAL was collected at necropsy from the right lung. The left lung was dissected prior to BAL collection and used for subsequent histo-pathology and virology procedures.

**RNA extraction**. RNA was isolated from nasal wash, throat swabs, EDTA treated whole blood, BAL and tissue samples (nasal cavity, tonsil, trachea, lung liver, jejunum and colon). Weighed tissue samples were homogenised and inactivated in RLT (Qiagen) supplemented with 1%(v/v) Beta-mercaptoethanol. Tissue

homogenate was then centrifuged through a QIAshredder homogenizer (Qiagen) and supplemented with ethanol as per manufacturer's instructions. Downstream extraction was then performed using the BioSprint™96 One-For-All vet kit (Qiagen) and Kingfisher Flex platform as per manufacturer's instructions. Non-tissue samples were inactivated in AVL (Qiagen) and ethanol, with final extraction using the QIAamp Viral RNA Minikit (Qiagen) as per manufacturer's instructions.

**Quantification of viral loads by RT-qPCR.** Reverse transcription-quantitative polymerase chain reaction (RT-qPCR) targeting a region of the SARS-CoV-2 nucleocapsid (N) gene was used to determine viral loads and was performed using TaqPath™ 1-Step RT-qPCR Master Mix, CG (Applied Biosystems™) and 2019-nCoV CDC RUO Kit (Integrated DNA Technologies). Sequences of the N1 primers and probe were: 2019-nCoV_N1-forward, 5′ GACCCCAAAATCAGCGAAAT 3′; 2019-nCoV_N1-reverse, 5′ TCTGGTTACTGCCAGTTGAATCTG 3′; 2019-nCoV_N1-probe, 5′ FAM-ACCCCGCATTACGTTTGGTGGACC-BHQ1 3′. Sequences can also be found in Supplementary Table 1. The cycling conditions were: 25 °C for 2 min, 50 °C for 15 min, 95 °C for 2 min, followed by 45 cycles of 95 °C for 3 s, 55 °C for 30 s. Tissues samples were tested on the QuantStudio™ 7 Flex Real-Time PCR System with an in vitro transcribed RNA standard of the full length SARS-CoV-2 N ORF (accession number NC_045512.2) with quantification between $1 \times 10^1$ and $1 \times 10^6$ copies/µl. Positive samples detected below the lower limit of quantification (LLOQ) were assigned the value of 5 copies/µl, whilst undetected samples were assigned the value of ≤2.3 copies/µl, equivalent to the assays lower limit of detection (LLOD). Non-tissues samples were tested on the 7500 Fast Real-Time PCR System (Applied Biosystems™) with a 100 bp Ultramer RNA oligo standard (Integrated DNA Technologies) equivalent to 28274-28373 bp of SARS-CoV-2 NC_045512.2, with quantification between $1 \times 10^1$ and $1 \times 10^7$ copies/µl. Positive samples detected below the limit of quantification were assigned the value of 6 copies/µl, whilst undetected samples were assigned the value of ≤2 copies/µl, equivalent to the assays LLOD.

**Subgenomic RT-qPCR.** Subgenomic RT-qPCR was performed on the Quant-Studio™ 7 Flex Real-Time PCR System using TaqMan™ Fast Virus 1-Step Master Mix (Thermo Fisher Scientific) and oligonucleotides as specified by Wolfel et al., with forward primer, probe and reverse primer at a final concentration of 250, 125 and 500 nM, respectively. Sequences of the sgE primers and probe were: 2019-nCoV_sgE-forward, 5′ CGATCTCTTGTAGATCTGTTCTC 3′; 2019-nCoV_sgE-reverse, 5′ ATATTGCAGCAGTACGCACACA 3′; 2019-nCoV_sgE-probe, 5′ FAM- ACACTAGCCATCCTTACTGCGCTTCG-BHQ1 3′. Sequences can also be found in Supplementary Table 1. Cycling conditions were 50 °C for 10 min, 95 °C for 2 min, followed by 45 cycles of 95 °C for 10 s and 60 °C for 30 s. RT-qPCR amplicons were quantified against an in vitro transcribed RNA standard of the full length SARS-CoV-2 E ORF (accession number NC_045512.2) preceded by the UTR leader sequence and putative E gene transcription regulatory sequence described by Wolfel et al. Positive samples detected below the LLOQ were assigned the value of 5 copies/µl, whilst undetected samples were assigned the value of ≤0.9 copies/µl, equivalent to the assays LLOD.

**SARS-CoV-2 virus plaque assay.** Samples were diluted in serum-free MEM containing antibiotic/antimycotic (Life Technologies) and incubated in 24-well plates (Nunc, ThermoFisher Scientific, Loughborough, UK) with Vero E6 cell monolayers. Virus was allowed to adsorb at 37 ºC for 1 h, then overlaid with MEM containing 1.5% carboxymethylcellulose (Sigma), 4% (v/v) foetal bovine serum (Sigma) and 25 mM HEPES buffer (Life Technologies). After incubation at 37 °C for 5 days, they were fixed overnight with 20% (w/v) formalin/PBS, washed with tap water and stained with methyl crystal violet solution (0.2% v/v) (Sigma).

**Plaque Reduction Neutralisation Test.** Neutralising virus titres were measured in heat-inactivated (56 °C for 30 min) serum samples. SARS-CoV-2 was diluted to a concentration of 933 pfu/ml (70 pfu/75 µl) and mixed 50:50 in 1% FCS/MEM with doubling serum dilutions from 1:10 to 1:320 in a 96-well V-bottomed plate. The plate was incubated at 37 °C in a humidified box for 1 h to allow the antibody in the serum samples to neutralise the virus. The neutralised virus was transferred into the wells of a washed plaque assay 24-well plate (see plaque assay method), allowed to adsorb at 37°C for a further hour, and overlaid with plaque assay overlay media. After 5 days incubation at 37 °C in a humified box, the plates were fixed, stained and plaques counted. Median neutralising titres (ND$_{50}$) were determined using the Spearman-Karber[44] formula relative to virus only control wells.

**Spike-specific IgG ELISA.** A full length trimeric and stabilised version of the SARS-CoV-2 Spike protein developed by LakePharma (TP30943F, construct #6) was used as antigen. High-binding 96-well plates (Nunc Maxisorp, 442404) were coated with 50 µl/well of 2 µg/ml Spike trimer in 1X PBS (Gibco) and incubated overnight at 4 °C. The ELISA plates were washed five times with wash buffer (1X PBS/0.05% Tween 20 (Sigma, P2287)) and blocked with 100 µl/well 5% Foetal Bovine Serum (FBS, Sigma, F9665) in 1X PBS/0.1% Tween 20 for 1 h at room temperature. After washing, serum samples were serially diluted in 10% FBS in 1X PBS/0.1% Tween 20, 50 µl/well of each dilution were added to the antigen coated plate and incubated for 2 h at room temperature. Following washing, anti-ferret IgG conjugated to HRP (Novus Biologics, NB7224) was diluted (1:5000) in 10% FBS in 1X PBS/0.1% Tween 20 and 100 µl/well were added to the plate, then incubated for 1 h at room temperature. After washing, 1 mg/ml O-Phenylenediamine dihydrochloride solution (Sigma P9187) was prepared and 100 µl/well were added. The development was stopped with 50 µl/well 1 M Hydrochloric acid (Fisher Chemical, J/4320/15) and the absorbance at 490 nm was read using Softmax 7.0. The cut-off was defined as the average Optical Density of naïve animals +2 standard deviation.

**Histopathological analysis.** Samples from the left cranial and left caudal lung lobe together with spleen, kidney, liver, tracheobronchial and axillary lymph nodes, jejunum, colon, trachea, larynx and nasal cavity, were fixed by immersion in 10% neutral-buffered formalin and processed routinely into paraffin wax. Nasal cavity samples were decalcified using an EDTA-based solution prior to embedding. 4 µm sections were cut and stained with haematoxylin and eosin (H&E) and examined microscopically. In addition, samples were stained using the RNAscope technique to identify the SARS-CoV-2 virus RNA. Briefly, tissues were pre-treated with hydrogen peroxide for 10 min (room temperature), target retrieval for 15 min (98–101 °C) and protease plus for 30 min (40 °C) (Advanced Cell Diagnostics). A V-nCoV2019-S probe (Cat No. 848561, Advanced Cell Diagnostics) was incubated on the tissues for 2 h at 40 °C. Amplification of the signal was carried out following the RNAscope protocol using the RNAscope 2.5 HD Detection kit – Red (Advanced Cell Diagnostics).

**Isolation of lung mononuclear cells.** Whole lungs were removed from each ferret. The lungs were dissected into small pieces and placed into a 12.5 ml solution of collagenase (715 collagenase units/ml) (Sigma-Aldrich) and DNase (350 DNase units/ml) (Sigma-Aldrich). Lungs were placed into gentleMACS C-tubes and agitated whilst incubating, 37 ºC for 1 h on an OctoMACS (Miltenyi Biotec, Surrey, UK). Partially digested lung tissue was then dissociated using an OctoMACS. The tissue solution was passed through two cell sieves (100 µm then 70 µm) and then layered with Ficoll®- Paque Premium (GE Healthcare, Hatfield, UK). Density gradient centrifugation was carried out at 400g for 30 min. Buffy coats containing lymphocytes were collected and washed with medium by pelleting cells via centrifugation at 400g for 10 min. The cells were counted using a Via1- cassette and a Nucleocounter-200 before cryopreservation in 95% FCS/5% v/v DMSO. Cryopreserved cells were then frozen at −80 °C in controlled rate freezer containers overnight, before transfer to liquid nitrogen (vapour phase).

**Interferon-gamma (IFN-γ) ELISpot assay.** An IFN-γ ELISpot assay was performed to determine the production capacity of SARS-CoV-2-specific T cells in the lung using a ferret IFN-γ kit (Mab-tech, Nacka. Sweden). Lung MNCs were defrosted into pre-warmed medium (R10) consisting of RPMI 1640 medium (Sigma-Aldrich) supplemented with 2 mM L-glutamine (Sigma-Aldrich), 0.05 mM 2-mercaptoethanol (Invitrogen, Paisley, UK), 25 mM HEPES buffer (Sigma-Aldrich, Dorset, UK), 100 U/ml Penicillin/100 µg/ml Streptomycin solution (Sigma-Aldrich), 10% heat inactivated foetal bovine serum (Sigma-Aldrich), and benzonase (Novogen, Merck, Darmstadt, Germany). Cells were rested for 2 h prior to use. Lung MNCs were assessed for responses to whole SARS-CoV-2 virus and a COVID-19 Spike Protein (GenBank: QHQ82464.1) peptide panel. The peptide panel consisted of 15 mer peptides overlapping by 11 mer. Individual peptides were reconstituted in 10% v/v DMSO. The 10 peptide pools, each containing 32 peptides, were created by combining equimolar amounts of each peptide. Three mega pools spanning the whole spike protein (approximately 100 peptides in each mega pool) were also created. Each peptide pool and mega pool was diluted for use in the ELISpot assay in supplemented RPMI to achieve a final concentration of 2.5 µg per peptide. SARS-CoV-2 whole virus was also used at an MOI of 0.09 to re-stimulate the lung MNCs. The virus was cell culture grown and was a direct match to the isolate used for ferret challenge. R10 media was used as a negative control and for preparation and dilution of cells, peptide, virus and stimulants. Cell stimulation cocktail (PMA/Ionomycin 500x concentrate, Sigma-Aldrich, Dorset, UK), was used as a positive control to prove cells were capable of a stimulation response. Pre-coated ferret anti- IFN-γ ELISpot plates (mAb MTF14, Mab-tech, Nacka. Sweden) were used and 500,000 lung MNCs were plated per well in 50 µl of R10, with or without antigen, in duplicate and incubated overnight (37 °C, 5% CO$_2$). Following cell stimulation, plates were washed 5x with 1x PBS (Gibco) and incubated at RT for 2 h with biotinylated anti IFN-γ IgG. The plates were then washed 5x with 1x PBS and incubated with streptavidin-ALP for 1 h, RT. The plates were washed again 5x with 1x PBS and spots were developed with 5-bromo-4-chloro-3-indoly phosphate (BCIP)-Nitro Blue tetrazolium (NBT) substrate. Plates were allowed to dry overnight and decontaminated by formaldehyde fumigation before removal from the CL3 facility. Plates were read, counted and quality control checked using the CTL ELISpot plate reader and ImmunoSpot 5.0 analyser software. Results from duplicate test wells were averaged. Data were corrected for background by subtracting the mean number of spots from the R10 media control wells from the mean counts of spots in the test wells.

**Reporting summary**. Further information on research design is available in the Nature Research Reporting Summary linked to this article.

## Data availability

A data source file is provided alongside this manuscript. The authors declare that the data supporting the findings of this study are available within the paper and its supplementary information files. Source data are provided with this paper.

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

## Acknowledgements

The authors gratefully acknowledge the support from the Biological Investigations Group at the National Infection Service, PHE, Porton Down, UK. The views expressed in this paper are those of the authors and not necessarily those of the funding body. This work was funded by the US Food and Drug Administration [Contract number: HHSF223201710194C].

## Author contributions

K.A.R., Y.H., S.G.F., C.J.W., J.A.H. and M.W.C. conceived the study. J.D. and M.C. provided virus strain. K.R.B. grew viral stock, optimised virology techniques and supervised virology experiments. S.A.F., D.J.H., I.T. and N.R.W. performed all animal procedures at containment level 3. K.A.R., P.B., B.E.C., K.G., C.M.K.H., T.B., O.S., D.N., K.S. and S.T. processed all animal samples at containment level 3. L.H., C.L.K., E.R. and F.J.S. contributed to histology experiments and performed critical assessment of pathology. L.A., E.B., K.R.B., N.S.C., K.J.G., H.E.H., S.L. and E.J.P. contributed wet virology experiments and analysis of data. K.A.R., L.A., E.B., S F-W., C.T., T.H., O.S., J.G., R.H., S.L., J.P., K.S. and N.I.W. contributed to inactivation, extraction and PCR of samples. G.S. performed quality control and analytical assistance on PCR data D.P.C., S.T.P. and K.L.O. performed NGS and analysis data. K.A.R. performed analysis on data generated. T.T., R.W. and M.J.D. provided technical assistance. K.A.R., J.A.T., F.J.S. and M.W.C. wrote the manuscript. A.C.M., C.J.W., K.S., B.E.C. and S.G.F. provided critical review.

## Competing interests

The authors declare no competing interests.
