## [Peer review file · Nature Communications]

Comments first round:

Reviewer #1 (Remarks to the Author):

The manuscript "Dose-dependent response to infection with SARS-CoV-2 in the ferret model: evidence of protection to re-challenge" describes the characterization of experimental SARS-CoV-2 infection in ferrets, as well as protection from re-challenge. While the ferret model for SARS-CoV-2 has been published by several groups, this study provides a more extensive characterization of the kinetics of infection. However, the authors limit the description to viral shedding clinical signs and pathology and do not include data on viral loads in the respiratory tract, or dissemination beyond.

General comments:

No data on infectious virus is provided in general.

No data on viral loads in tissues is provided.

Many data points are based on n=1, which does not allow comparison or statistical analysis between groups or over time, especially since these are outbred animals and therefore animal-to-animal variation is expected.

The discussion should be shortened, the comparison between the other ferret studies can be condensed. The authors should include a section on the benefit of ferret models over hamster or NHP models.

Specific comments:

Line 109, there already are several published studies on SARS-CoV-2 infection in ferrets (Kim et al, Shi et al, park et al), please reference these.

Line 111, from published literature it seems they are suitable, please rephrase.

Line 122, 1mL IN seems like a high volume, what is the rationale for this?

Line 122, how was the age of the animals selected? Given that advanced age is associated with increased disease severity, why were experiments performed with relatively young animals?

Line 128, why were animals from group 1 not re-challenged?

Line 133, in addition to viral RNA detection, please include data on shedding of infectious virus as this is more relevant.

Line 200, in addition to viral shedding, please provide viral loads RNA and/or infectious virus in tissues to characterize tissue tropism along with cell tropism.

Line 219, did viral RNA positive cells correlate with presence of histopathological lesions?

Line 224, was there evidence of fibrosis in lungs on day 21?

Line 239, please show data on the kinetics of antibody responses for individual animals in a graph, rather than a table. In addition to neutralizing antibody titers, please also include data on SARS-CoV-2 specific IgM, IgA and IgG responses.

Line 239, given the recent reports of natural SARS-CoV-2 infection in mink and the presence of other veterinary coronaviruses in different species, were animals screened for presence of SARS-CoV-2 reactive antibodies? And if so, were any of them seropositive at the start of the study?

Line 243, did all low dose animals seroconvert as evidence of infection in absence of viral RNA shedding?

Line 250, so this control animal has accidentally become infected during the experiment?

Line 282, from the legend it seems this is based on n=1, if so, you cannot correlate response with treatment group.

Line 309, so no evidence of seroconversion in the other animals with low dose?

Line 342, given the presence of shedding from rectal swabs, was there evidence of infection in the alimentary tract?

Line 362, please provide data on infectious virus.

Line 369, have you tried detecting subgenomic RNA as evidence of active replication?

Line 370, this is usually later in infection when neutralizing antibodies may be present in the sample, which is not the case early in your study.

Line 374, please provide neutralizing titers at day 0 to show that day 8 titers are actually due to challenge.

Line 401, the data suggests euthanizing between day 3 and 7 pc for optimal lung pathology, on

what data do you base day 14 pc?

Line 492, what tissues were collected during necropsy?

Line 498, what virological procedures were performed on lung tissue and where is this shown?

Line 507, in order to make data more comparable with human data, the E and RdRp based assays described by Corman et al.

Reviewer #2 (Remarks to the Author):

The article, "Dose-dependent response to infection with SARS-CoV-2 in the ferret model: evidence to protection to re-challenge", seeks to characterize the most suitable challenge dose, understanding the kinetics of viral pathogenesis and immune responses following SARS-CoV-2 infection.

The article, when taken as a whole, is of relevance and which focuses on the current issues of the ongoing pandemic. While the study has merit, the methods should be re-evaluated prior to consideration for publication in this journal.

There are at least 3 major concerns this reviewer found on this study:

1. number of sacrificed ferret for each treatment group on a certain time period gives questionable statistics, number should be minimum of 3 to demonstrate a statistically significant and meaningful differences. Small sample size doesn't balance out factors as well as eliminating possible biases that could occur. The power of the study could be improved by obtaining a larger sample size or repeating the test.
2. It would also be helpful if data regarding the viability of the virus were added. Virus isolation from different samples collected would give more concrete evidence on the presence of shedding of viable SARS-CoV-2 from collected specimens. Without these, it is insubstantial to evaluate viral shedding post challenge.
3. One of naïve ferrets (n=2, group 4 and 5) showing no clinical signs and were negative on SARS-CoV-2 PCR detection (p.12) and showed strong neutralizing antibodies- 120 (p.38 table 3) in both primary and re-infection periods. How these things are happening? Are they were transmitted from infection animal in BSL3 facility or they were originally infected prior to infection. At any case, these results may affect the reliability of other results. It should be clarified.

Reviewer #3 (Remarks to the Author):

This is an interesting manuscript that examines the effect of different SARS-CoV-2 doses on disease in ferrets. Very few SARS-CoV-2 ferrets infection studies have been published thus far, and ferrets have not been used very much in wider coronavirus research, so this manuscript could make an important contribution to the field.

1. The evaluation of viral shedding in infected animals and clinical observations are mostly well done. However, the "cumulative incidence" (shown in Figure 2a) and clinical scores shown in Table 2 are very confusing. Table 2 shows summed individual instances of clinical parameters across all animals in a group during a defined time interval. In Figure 2a this appears to be broken down to show those aggregated data for individual days. It would be much better to show the clinical scores for individual animals so that what is shown in the table corresponds to the graph.
2. What is meant by "ruffled fur"? This is a common criterion for clinical evaluation of mice, which groom frequently. Ferrets, on the other hand, do not groom to any discernible extent, so it difficult to imagine how ruffled fur could be meaningfully evaluated. Pictures of "normal" and "ruffled" fur on ferrets should be shown so that the difference is clear.
3. Were other clinical parameters evaluated? As an upper respiratory infection, SARS-CoV-2 may cause increased mucous secretion, nasal congestion, and wheezing. Were any such signs noted? If so, the data should be shown and discussed.
4. Fever $>39.9^{\circ}\text{C}$ is a very high threshold. Why was it chosen? Other respiratory infections can induce temperatures between 39 and 40°C , which would still be classified as fever.
5. Figure 3: Larger images of the areas indicated by the arrows and asterisks should be shown in

the figure to make interpretation easier. The images are rather small and fine details are difficult to see. Negative control panels from non-infected ferret tissues should also be shown for comparison.

6. Figure 4: There is reduced shedding of virus in animals after re-challenge, but it still quite high (10^{5-6} genome copies /ml). Likewise, the re-challenged animals all lost some weight, while the naïve animals did not, and none of the animals in the initial infection lost any weight. This rather supports the conclusion that re-infection with SARS-CoV-2 likely leads to a more severe clinical course, not that there is significant protection. The alternative explanation for this increased disease severity offered in the Discussion (see page 19, lines 389-391) is not at all believable. With such high neutralizing titers in the Medium Group (PRNT₅₀ >260), it is quite surprising that virus replication in the nose is so high. This should be discussed in more depth. It would benefit the manuscript to discuss how neutralizing titers can be quite high, but that SARS-CoV-2 shedding is only moderately affected.

7. Given the evidence of more severe clinical disease upon re-infection, the authors should comment on what potential role processes such as antibody-dependent enhancement might play.

NCOMMS-20-20839 Response to Reviewers

Thank you for the return of our manuscript. We are pleased that the reviewers recognised the importance of our work. Following is our detailed response to the various points raised by each reviewer.

REVIEWER COMMENTS

Reviewer #1 (Remarks to the Author):

The manuscript “Dose-dependent response to infection with SARS-CoV-2 in the ferret model: evidence of protection to re-challenge” describes the characterization of experimental SARS-CoV-2 infection in ferrets, as well as protection from re-challenge. While the ferret model for SARS-CoV-2 has been published by several groups, this study provides a more extensive characterization of the kinetics of infection. However, the authors limit the description to viral shedding clinical signs and pathology and do not include data on viral loads in the respiratory tract, or dissemination beyond.

General comments:

No data on infectious virus is provided in general.

As stated in the original manuscript due to technical issues in how samples were processed (line 411) and sensitivity of our assay we were unable to consistently detect infectious virus in many of the ferret samples. However, we have repeated our attempts to detect infectious virus and samples that were found to be positive have now been highlighted accordingly on a modified Figure 1.

Additionally, to illustrate we are detecting virus replication in our model we have performed sub-genomic PCR for throat swab, BAL and tissue samples (this new data has been added Figure 1, Figure 2 and Supplementary Figure 5).

No data on viral loads in tissues is provided.

We agree this is a gap in our data set and qPCR has now been carried out on nasal cavity, trachea, tonsil, lung, liver, jejunum and colon. Results have been included in manuscript with additional figures (new Figure 2 and Supplementary Figure 1).

Many data points are based on n=1, which does not allow comparison or statistical analysis between groups or over time, especially since these are outbred animals and therefor animal-to-animal variation is expected.

Yes, we agree. However, this is an initial dose ranging study and as per several other published early reports minimal numbers of animals have been used for sequential culls. Additionally, taken together there are at least 2 positive animals from the high and medium dose groups for each time

point.

The discussion should be shortened, the comparison between the other ferret studies can be condensed. The authors should include a section on the benefit of ferret models over hamster or NHP models.

The discussion has been re-worked to condense the comparison between other ferret studies and a comparison of the benefits of the ferret over NHPs and hamsters has been included. NHPs are only used when no other suitable model is available due to ethical reasons. Hamsters provide a good model for increased pathology and symptoms but cannot be serially sampled. The ferret has larger blood volume and recently T cell analysis reagents have become available. It has also been extensively characterised for other important human respiratory viruses e.g. influenza. Therefore, the animal model chosen would depend on the question being asked. For example, ferrets may provide a better model for co-infection and vaccine studies and hamsters optimal for therapeutics.

Specific comments:

Line 109, there already are several published studies on SARS-CoV-2 infection in ferrets (Kim et al, Shi et al, park et al), please reference these.

These have been referenced, furthermore these studies are discussed and referenced in the discussion.

Line 111, from published literature it seems they are suitable, please rephrase.

We agree that ferrets appear to be a suitable model. However, the dose ranging was performed to ascertain the most suitable dose. Importantly a dose ranging study of this nature has not yet been published.

Line 122, 1mL IN seems like a high volume, what is the rationale for this?

The volume was chosen as it is known from prior studies with influenza when using 1ml a percentage of challenge material will reach the lungs. This rationale has been added to the manuscript (line 122).

Line 122, how was the age of the animals selected? Given that advanced age is associated with increased disease severity, why were experiments performed with relatively young animals?

Generally, only minimal numbers of older animals are available from breeding colonies. Limited supplies would mean subsequent studies on COVID interventions would not be possible. The aim of our study was to characterise a model for the evaluation of interventions.

Line 128, why were animals from group 1 not re-challenged?

The decision was made to cull the two remaining high dose ferrets at day 21, and therefore not re-challenge them, in the hope to get more information regarding virology and pathology at the later time point.

Line 133, in addition to viral RNA detection, please include data on shedding of infectious virus as this is more relevant.

As stated in the original manuscript due to technical issues in sensitivity of our assay we were unable to consistently detect infectious virus in many of the ferret samples. However, we have repeated our attempts to detect infectious virus and samples that were found to be positive have now been highlighted accordingly on a modified Figure 1.

Additionally, to illustrate we are detecting virus replication in our model we have performed sub-genomic PCR for throat swab, BAL and tissue samples (this new data has been added to Figure 1 and a new Figure 2 has been added).

Line 200, in addition to viral shedding, please provide viral loads RNA and/or infectious virus in tissues to characterize tissue tropism along with cell tropism.

qPCR has now been carried out on nasal cavity, trachea, tonsil, lung, liver, jejunum and colon. Results have been included in manuscript with additional figure. Results have been incorporated into a new figure, Figure 2, and have been discussed.

Line 219, did viral RNA positive cells correlate with presence of histopathological lesions?

No, only scattered cells in areas without lesions were observed. No viral RNA was detected in the bronchiolar infiltrates from the sections analysed.

Line 224, was there evidence of fibrosis in lungs on day 21?

There was no evidence of fibrosis in the lungs at day 21 post challenge.

Line 239, please show data on the kinetics of antibody responses for individual animals in a graph, rather than a table. In addition to neutralizing antibody titers, please also include data on SARS-CoV-2 specific IgM, IgA and IgG responses.

Additional assays to identify SARS-CoV-2 Spike specific IgG have now been carried out and results added to the paper as requested (Supplementary Figure 2).

Line 239, given the recent reports of natural SARS-CoV-2 infection in mink and the presence of other veterinary coronaviruses in different species, were animals screened for presence of SARS-CoV-2 reactive antibodies?

Yes, this data is available now and included.

We know from previous work with this outbred ferret colony, that much like humans, ferrets can be infected by common circulating coronaviruses. Ferret nasal washes were screened one day prior to infection. Two animals were found to be positive for alphacoronavirus sequence at this time point, their baseline IgG and ND50 titres were not found to be abnormally high when compared to the other ferrets sampled.

And if so, were any of them seropositive at the start of the study?

The ferrets that were used in this study were outbred and as expected the ND₅₀ values and SARS-CoV-2 Spike specific IgG titres were variable between ferrets. However, those ferrets that were successfully infected clearly show an increase in ND₅₀(PRNT) and SARS-CoV-2 Spike specific IgG titres (ELISA) over baseline titres.

Line 243, did all low dose animals seroconvert as evidence of infection in absence of viral RNA shedding?

Five out of six ferrets in the low dose group appeared not to seroconvert. This is based on the SARS-CoV-2 Spike specific IgG titres. The low dose ferret culled at 14 days post challenge (dpc) had increased endpoint titres at 14 dpc. However, the two ferrets that were re-challenged did not appear to seroconvert following the original challenge.

Line 250, so this control animal has accidentally become infected during the experiment?

We believe that this could be likely. Naïve PBS animals were housed in the same CL3 facility as infected ferrets. They were housed in a separate cage and handled prior to handling of SARS-CoV-2 infected ferrets. The additional SARS-CoV-2 Spike specific IgG ELISA that has been carried out suggests that this animal may have seroconverted.

Line 282, from the legend it seems this is based on n=1, if so, you cannot correlate response with treatment group.

We're not attempting to correlate a response with treatment group as we agree this cannot be done with a n=1.

Line 309, so no evidence of seroconversion in the other animals with low dose?

As previously stated, five out of six ferrets appeared not to seroconvert using our IgG Spike ELISA.

Line 342, given the presence of shedding from rectal swabs, was there evidence of infection in the alimentary tract?

qPCR shows the absence of viral RNA in jejunum and colon during the first challenge.

However, scattered cells (absorbing epithelial and Goblet cells) positive by RNAscope ISH were detected in both small and large intestine but were not associated with any lesion.

Line 362, please provide data on infectious virus.

As stated in the original manuscript due to technical issues in how samples were processed (line 411) and sensitivity of our assay we were unable to consistently detect infectious virus in many of the ferret samples. However, we have repeated our attempts to detect infectious virus and samples that were found to be positive have now been highlighted accordingly on a modified

Figure 1. Samples which gave infectious virus in the plaque assay have been highlighted on Figure 1 with filled symbols.

Line 369, have you tried detecting subgenomic RNA as evidence of active replication?

As stated previously we have now performed qPCR to detect subgenomic RNAd. This additional data has been included in a new Figure 1 and Figure 2.

Line 370, this is usually later in infection when neutralizing antibodies may be present in the sample, which is not the case early in your study.

Yes we agree.

Line 374, please provide neutralizing titers at day 0 to show that day 8 titers are actually due to challenge.

Neutralising titres at -1 days post challenge have now been provided.

Line 401, the data suggests euthanizing between day 3 and 7 pc for optimal lung pathology, on what data do you base day 14 pc?

Animals were culled at day 14 to see if there were any longer lasting lung pathology caused by the virus.

Line 492, what tissues were collected during necropsy?

Details of tissue collected have been added to the "Necropsy Procedures" (line 508).

Line 498, what virological procedures were performed on lung tissue and where is this shown?

qPCR was carried out on tissues. These results have been added to the manuscript.

Line 507, in order to make data more comparable with human data, the E and RdRp based assays described by Corman et al.

The CDC assay is an accepted qPCR to determine the genome copy numbers. We believe general ranges can be compared with human data that employed alternative qPCR methodologies.

Reviewer #2 (Remarks to the Author):

The article, "Dose-dependent response to infection with SARS-CoV-2 in the ferret model: evidence to protection to re-challenge", seeks to characterize the most suitable challenge dose, understanding the kinetics of viral pathogenesis and immune responses following SARS-CoV-2 infection.

The article, when taken as a whole, is of relevance and which focuses on the current issues of the ongoing pandemic. While the study has merit, the methods should be re-evaluated prior to consideration for publication in this journal.

There are at least 3 major concerns this reviewer found on this study:

1. number of sacrificed ferret for each treatment group on a certain time period gives questionable statistics, number should be minimum of 3 to demonstrate a statistically significant and meaningful differences. Small sample size doesn't balance out factors as well as eliminating possible biases that could occur. The power of the study could be improved by obtaining a larger sample size or repeating the test.

Yes, we completely agree. However, as previously stated, this is an initial dose ranging study and as per several other early reports minimal numbers of animals have been used for sequential culls. Additionally, taken together there are at least 2 positive animals from the high and medium dose groups for each time point.

2. It would also be helpful if data regarding the viability of the virus were added. Virus isolation from different samples collected would give more concrete evidence on the presence of shedding of viable SARS-CoV-2 from collected specimens. Without these, it is insubstantial to evaluate viral shedding post challenge.

As stated in the original manuscript due to technical issues in how samples were processed (line 411) and sensitivity of our assay we were unable to consistently detect infectious virus in many of the ferret samples. However, we have repeated our attempts to detect infectious virus and samples that were found to be positive have now been highlighted accordingly on a modified Figure 1. Samples which gave infectious virus in the plaque assay have been highlighted on Figure 1 with filled symbols.

Additionally, to illustrate we are detecting virus replication in our model we have performed sub-genomic PCR for throat swab, BAL and tissue samples (this new data has been added to Supplementary Figure 1 and Supplementary Figure 2).

3. One of naïve ferrets (n=2, group 4 and 5) showing no clinical signs and were negative on SARS-CoV-2 PCR detection (p.12) and showed strong neutralizing antibodies- 120 (p.38 table 3) in both primary and re-infection periods. How these things are happening? Are they were transmitted from infection animal in BSL3 facility or they were originally infected prior to infection. At any case, these results may affect the reliability of other results. It should be clarified.

We believe that one of the ferret from group 4 could have become infected during the course of the first challenge phase. The ferrets were housed in the same CL3 facility in separate cages from animals that were intentionally infected. The two naïve sentinel ferrets were house in a pair together. The additional SARS-CoV-2 Spike specific IgG ELISA that has been carried out suggests that the animal this animal may have seroconverted.

Reviewer #3 (Remarks to the Author):

This is an interesting manuscript that examines the effect of different SARS-CoV-2 doses on disease in ferrets. Very few SARS-CoV-2 ferrets infection studies have been published thus far, and ferrets

have not been used very much in wider coronavirus research, so this manuscript could make an important contribution to the field.

1. The evaluation of viral shedding in infected animals and clinical observations are mostly well done. However, the “cumulative incidence” (shown in Figure 2a) and clinical scores shown in Table 2 are very confusing. Table 2 shows summed individual instances of clinical parameters across all animals in a group during a defined time interval. In Figure 2a this appears to be broken down to show those aggregated data for individual days. It would be much better to show the clinical scores for individual animals so that what is shown in the table corresponds to the graph.

The figure has now been changed to record the number of instances at each time point despite the number of ferrets so the graph now corresponds better to the table.

2. What is meant by “ruffled fur”? This is a common criterion for clinical evaluation of mice, which groom frequently. Ferrets, on the other hand, do not groom to any discernible extent, so it difficult to imagine how ruffled fur could be meaningfully evaluated. Pictures of “normal” and “ruffled” fur on ferrets should be shown so that the difference is clear.

This is a very good point. However, ruffled fur is described in veterinary practice for a variety of species, including dogs, cats and ferrets. While we agree with the grooming comment, we can see signs of poor health in all animal species by looking at the condition of the coat. Our experienced animal technicians commonly use this criteria in clinical scoring.

3. Were other clinical parameters evaluated? As an upper respiratory infection, SARS-CoV-2 may cause increased mucous secretion, nasal congestion, and wheezing. Were any such signs noted? If so, the data should be shown and discussed.

These clinical parameters were looked for during the infection but did not become apparent in any of the infected ferrets.

4. Fever $>39.9^{\circ}\text{C}$ is a very high threshold. Why was it chosen? Other respiratory infections can induce temperatures between 39 and 40°C , which would still be classified as fever.

In our experience ferrets can have a normal daily temperature range from 37.0 to 39.8°C. Temperatures have been reassessed for each individual ferret. Using the baseline temperatures of all of the ferrets used in this study an average baseline temperature of 38.3°C was calculated. A fever was then defined as $\pm 1^{\circ}\text{C}$ from baseline recorded at least two consecutive occasions post-challenge. After analysis we found that none of the ferrets experienced a fever during the post challenge phase.

5. Figure 3: Larger images of the areas indicated by the arrows and asterisks should be shown in the figure to make interpretation easier. The images are rather small and fine details are difficult to see. Negative control panels from non-infected ferret tissues should also be shown for comparison.

A larger, clearer panel has been prepared for Figure 3 (now figure 4).

6. Figure 4: There is reduced shedding of virus in animals after re-challenge, but it still quite high (10^5 -

10⁶ genome copies /ml). Likewise, the re-challenged animals all lost some weight, while the naïve animals did not, and none of the animals in the initial infection lost any weight. This rather supports the conclusion that re-infection with SARS-CoV-2 likely leads to a more severe clinical course, not that there is significant protection. The alternative explanation for this increased disease severity offered in the Discussion (see page 19, lines 389-391) is not at all believable. With such high neutralizing titers in the Medium Group (PRNT50 >260), it is quite surprising that virus replication in the nose is so high. This should be discussed in more depth. It would benefit the manuscript to discuss how neutralizing titers can be quite high, but that SARS-CoV-2 shedding is only moderately affected.

The disease presented by the re-challenged ferrets is still very mild, and while there is a suggestion of enhanced disease it cannot be classified as severe. We re-challenged ferrets with the high dose of 5x10⁶ pfu/ml, so it isn't unusual that initial transient viral replication in the nose following re-challenge appears to be so high despite high neutralising antibody titres.

7. Given the evidence of more severe clinical disease upon re-infection, the authors should comment on what potential role processes such as antibody-dependent enhancement might play.

We are not suggesting that we are seeing ADE on re-infection. Moreover, we only describe mild clinical signs.

Comments second round:

Reviewer #2 (Remarks to the Author):

NCOMMS-20-20839A

Overall the manuscript is much improved from its initial version.

However, still, not enough number of animals were adequately tested for statistical analysis.

Therefore, statistical analysis cannot be conducted.

Although the author described the possible explanation for low viable virus isolation from their specimens at line 422 to 432, it is hard to agree with. Most of the clinical specimen collection methods used are similar to other manuscripts, but still, the viable virus isolation rate was quite low compared with cited references. These might be associated with other factors, such as the duration times from collection to virus infection in cells, testing cell lines, and the number of freezing and thawing times. Further, this study used younger ferrets (7-months old) compared with other studies (some study used more than 12 moth old animals). The referenced manuscripts (Shi et al., Kim et al., and Richard et al.) demonstrated a rapid and high incidence of direct contact transmissions suggesting that there were enough infectious viruses in the upper respiratory tracts in ferrets.

As an alternative method to measure the replicating viruses in animals, the author tested sgRNA PCR methods. Why was there a huge difference between N-gene based qRT-PCR and E gene-based sgRT-PCR? These differences in copy number may be correlated with cell-free infectious virus virions. If this is the case, the virus isolation rate might be much higher than tested in this study. One more point, what exactly is the meaning of "viral RNA was detectable but unquantifiable" in many places. One of the main reasons to conduct the qRT-PCR is the quantification of virus copy number. If this unquantifiable represents the status of upper or lower levels of the Standard curve of known virus titers, it should be considered more carefully whether it is truly positive or negative.

Reviewer #3 (Remarks to the Author):

The authors have done a credible job of addressing the points raised in the reviews and the revised manuscript is much improved. Given the absence of lung pathology in re-challenged ferrets one could reasonably claim some protection from re-infection despite no obvious differences in upper respiratory tract replication. The disease is still very mild in ferrets so this is difficult to assess and such claims should still be treated with caution. However, no major concerns remain.

Response to reviewers

NCOMMS-20-20839A

Thank you for acceptance of our manuscript. Please see below for our response to additional comments from the reviewers.

Reviewer #2 (Remarks to the Author):

Overall the manuscript is much improved from its initial version. However, still, not enough number of animals were adequately tested for statistical analysis. Therefore, statistical analysis cannot be conducted.

Agreed, statistical analysis cannot be conducted due to the low number of animals used.

Although the author described the possible explanation for low viable virus isolation from their specimens at line 422 to 432, it is hard to agree with. Most of the clinical specimen collection methods used are similar to other manuscripts, but still, the viable virus isolation rate was quite low compared with cited references.

These might be associated with other factors, such as the duration times from collection to virus infection in cells, testing cell lines, and the number of freezing and thawing times. Further, this study used younger ferrets (7-months old) compared with other studies (some study used more than 12 moth old animals). The referenced manuscripts (Shi et al., Kim et al., and Richard et al.) demonstrated a rapid and high incidence of direct contact transmissions suggesting that there were enough infectious viruses in the upper respiratory tracts in ferrets.

As an alternative method to measure the replicating viruses in animals, the author tested sgRNA PCR methods.

One more point, what exactly is the meaning of “viral RNA was detectable but unquantifiable” in many places. One of the main reasons to conduct the qRT-PCR is the quantification of virus copy number. If this unquantifiable represents the status of upper or lower levels of the Standard curve of known virus titers, it should be considered more carefully whether it is truly positive or negative.

Lower limits of detection and quantification have been set as described in the methods and illustrated on the graphs. To ensure that it is clear that these results fall below the quantifiable level of the assay the wording of the text has been changed to state that viral RNA was detected above the level of detection of the assay but below the level of quantitation.

Reviewer #3 (Remarks to the Author):

The authors have done a credible job of addressing the points raised in the reviews and the revised manuscript is much improved. Given the absence of lung pathology in re-challenged ferrets one could reasonably claim some protection from re-infection despite no obvious differences in upper respiratory tract replication. The disease is still very mild in ferrets so this is difficult to assess and such claims should still be treated with caution. However, no major concerns remain.

In line 449 we have highlighted that protective immunity findings, after re-challenge, are based on a small number of animals.